# Correlation between Neutrophil Extracellular Traps (NETs) Expression and Primary Graft Dysfunction Following Human Lung Transplantation

**DOI:** 10.3390/cells11213420

**Published:** 2022-10-29

**Authors:** Steven Bonneau, Caroline Landry, Stéphanie Bégin, Damien Adam, Louis Villeneuve, Marie-Élaine Clavet-Lanthier, Ariane Dasilva, Elcha Charles, Benjamin L. Dumont, Paul-Eduard Neagoe, Emmanuelle Brochiero, Ahmed Menaouar, Basil Nasir, Louis-Mathieu Stevens, Pasquale Ferraro, Nicolas Noiseux, Martin G. Sirois

**Affiliations:** 1Research Center—Montreal Heart Institute, 5000 Belanger St., Montreal, QC H1T 1C8, Canada; 2Centre de Recherche du Centre Hospitalier de l’Université de Montréal (CRCHUM), 900 Saint-Denis St, Montreal, QC H2X 0A9, Canada; 3Department of Medicine, Faculty of Medicine, Université de Montréal, 2900 Blvd Édouard-Montpetit, Montreal, QC H3T 1J4, Canada; 4Department of Pharmacology and Physiology, Faculty of Medicine, Université de Montréal, 2900 Blvd Édouard-Montpetit, Montreal, QC H3T 1J4, Canada

**Keywords:** lung transplant, primary graft dysfunction, neutrophils, NETs, cytokines

## Abstract

Primary graft dysfunction (PGD) is characterized by alveolar epithelial and vascular endothelial damage and inflammation, lung edema and hypoxemia. Up to one-third of recipients develop the most severe form of PGD (Grade 3; PGD3). Animal studies suggest that neutrophils contribute to the inflammatory process through neutrophil extracellular traps (NETs) release (NETosis). NETs are composed of DNA filaments decorated with granular proteins contributing to vascular occlusion associated with PGD. The main objective was to correlate NETosis in PGD3 (n = 9) versus non-PGD3 (n = 27) recipients in an exploratory study. Clinical data and blood samples were collected from donors and recipients pre-, intra- and postoperatively (up to 72 h). Inflammatory inducers of NETs’ release (IL-8, IL-6 and C-reactive protein [CRP]) and components (myeloperoxidase [MPO], MPO-DNA complexes and cell-free DNA [cfDNA]) were quantified by ELISA. When available, histology, immunohistochemistry and immunofluorescence techniques were performed on lung biopsies from donor grafts collected during the surgery to evaluate the presence of activated neutrophils and NETs. Lung biopsies from donor grafts collected during transplantation presented various degrees of vascular occlusion including neutrophils undergoing NETosis. Additionally, in recipients intra- and postoperatively, circulating inflammatory (IL-6, IL-8) and NETosis biomarkers (MPO-DNA, MPO, cfDNA) were up to 4-fold higher in PGD3 recipients compared to non-PGD3 (*p* = 0.041 to 0.001). In summary, perioperative elevation of NETosis biomarkers is associated with PGD3 following human lung transplantation and these biomarkers might serve to identify recipients at risk of PGD3 and initiate preventive therapies.

## 1. Introduction

Primary graft dysfunction (PGD) is the leading cause of morbidity and mortality in the perioperative period of lung transplantation, characterized by vascular endothelial and alveolar epithelial damage and inflammation, lung edema and severe hypoxemia [1]. The severity of PGD is graded with the PaO_2_/FiO_2_ ratio and edema infiltrates at chest X-ray, where a PaO_2_/FiO_2_ ratio of 200 mmHg or less defines the worst PGD score (e.g., PGD3) [2]. Unfortunately, one third of recipients develop PGD3 within 72 h post-transplantation despite providing efforts to enhance donor lung suitability, organ preservation and perioperative management [2,3,4]. The 30-day mortality of PGD3 recipients is up to 36% [4,5,6,7,8]. Until now, there is neither an effective method predicting which donor lungs will develop PGD3, nor a successful therapy for PGD, the management being only supportive [3].

Neutrophils play a key role in PGD [4,9,10]. Their detrimental implication is rooted into ischemia-reperfusion injury (IRI), a process considered the leading culprit of PGD [11,12,13,14]. In animal models of lung IRI, neutrophils are sequestered in the pulmonary microcirculation during ischemia, and further recruited following reperfusion [9,15,16,17,18]. In humans, priming and activation of circulating neutrophils is likely to occur, given the highly pro-inflammatory environment in donors and recipients [4,5,9,19,20]. Donor and recipient levels of two cytokines involved in neutrophilic chemotaxis or migration (IL-8 and IL-6) are potent predictors of PGD development [21,22,23,24,25]. Thus, IRI and systemic inflammation can provide a suitable environment for neutrophil activation and neutrophil extracellular traps (NETs) formation [9,11].

NETs are web-like structures composed of decondensed chromatin, histones and antimicrobial proteins such as myeloperoxidase (MPO) and neutrophil elastase (NE) [26]. Originally discovered for their role against bacterial infections [26], NETs are also key drivers of several diseases. In lung transplantation, the role of circulating NETs is less clear. Increased NETs in perfusates from ex vivo lung perfusion (EVLP) during donor lung reconditioning was associated with worse recipient outcome [27]. In a murine model of PGD, Sayah etal. [10] demonstrated a platelet-dependent formation of NETs and the benefits of inhaled DNase I therapy by reducing pulmonary lesions. They also demonstrated the pathological implication of NETs in human airways but found no difference in plasma levels of NETs between PGD3 and PGD0 human recipients. However, plasma samples were only collected from recipients and at limited time-points. Thus, there is a need to characterize the kinetics of circulating NETs to identify patients at risk of PGD in order to better anticipate and prevent lethal complications.

Our objectives were to determine the pre-, intra- and postoperative levels of NETs and cytokines associated with NETosis in donors and recipients undergoing bilateral lung transplantation. Cytokines and NETs were correlated with the development of PGD3 in recipients. We hypothesized that pulmonary NETosis would be highly pathological, leading to endothelial injury and microvascular occlusion via neutrophil/platelets aggregates adhered onto endothelial cells. We posed that elevation of acute markers of inflammation in both donors and recipients provides a suitable milieu for NETs synthesis and PGD development.

## 2. Materials and Methods

### 2.1. Participants

This observational study was conducted from June 2018 to December 2020 at the Centre hospitalier de l’Université de Montréal (CHUM), Québec, Canada. Exclusion criteria were re-transplant procedures, use of extracorporeal membrane oxygenation (ECMO) before surgery, patients with tuberculosis, HIV or hepatitis. Recipients undergoing cardiopulmonary bypass (CPB) or ECMO intra- or post-surgery were included in the study. These similar supportive procedures are commonly used in cardiothoracic surgery to support systemic perfusion and/or blood oxygenation. CPB is only used intraoperatively while ECMO can be used in surgery as well as in intensive care units due to its smaller size and mobility.

### 2.2. Study Design

Venous blood samples were obtained from donors (n = 25) and recipients (n = 36) in Vacutainer separation tubes containing protamine (1 mg/mL) to ensure coagulation. In donors, blood was obtained during surgery before lung procurement. In recipients, blood was collected after general anesthesia induction, during surgery before the first lung implantation and after the second lung engraftment, and at 3, 24, 48 and 72 h following second lung reperfusion. Samples were centrifuged (1500× *g* for 15 min) and serum was aliquoted and frozen at −80 °C.

Demographics and clinical parameters of donors and recipients were collected from the electronic medical record. Clinical data from all corresponding donors were included in the recipients’ medical records. The outcome was a binary variable defined by PGD3 development at 12 h following arrival of recipients in ICU post-transplant, a cut-off previously used in other studies (between 0 to 12 h post-ICU arrival) [28,29,30]. This allows to identify earlier the most severely affected patients (PGD3) and separate them from patients without PGD (PGD0) or low-PGD (PGD1/2) grade. Edema infiltrates on chest X-ray and PaO_2_/FiO_2_ ratios were used according to the 2016 standardized scale of the International Society for Heart and Lung Transplantation (ISHLT) to determine PGD grades [2].

### 2.3. Quantification of Circulating NETs and Biomarkers

As NETs are composed of DNA strands bound with proteins, including MPO, a custom ELISA using a mouse anti-human MPO capture antibody and an anti-DNA from the Cell Death Detection ELISA (Sigma-Aldrich, Oakville, ON, Canada) as detection antibody was performed [31]. Cell-free DNA was isolated with QIAamp^®^ Blood Mini Kit (#51106, Qiagen, Valencia, CA, USA) and quantified with Quant-IT PicoGreen dsDNA Assay Kit (#P7589; Invitrogen, Eugene, OR, USA). MPO was measured by ELISA (#440007, Biolegend, San Diego, CA, USA). IL-8 and IL-6 were quantified by a high sensitivity multiplex assay (Luminex Assay; #LHSCM206 and #LHSCM208, R&D Systems, Minneapolis, MN, USA). High-sensitivity CRP (hsCRP) was quantified by nephelometry at the MHI clinical biochemistry laboratory.

### 2.4. Lung Histology, Immunohistology and Confocal Imaging

As per standard procurement technique, donor lungs were perfused anterogradely (4 L, pulmonary artery) and retrogradely (1 L, pulmonary veins) with a low-potassium dextran solution (Perfadex^®^, XVIVO Perfusion, Gothenburg, Sweden) [32]. In cases of size mismatch between donor lungs and recipients, lung specimens were obtained after lung volume reduction (peripheral segmental resection) or standard lobectomy in the donor lung. These procedures were performed during lung transplantation before lung reperfusion in the recipient. Collected tissues (1 specimen of ~2 cm^3^ in volume per available graft) were put into 10% neutral-buffered formalin solution and embedded in paraffin blocks. Neutrophils were detected using an anti-myeloperoxidase (MPO, #Pa5-16672; Thermo Fisher, Waltham, MA, USA) and NETosis was visualized with an anti-histone H3 citrulline R2 + R8 + R17 (CitH3, #ab5103, Abcam, Toronto, ON, Canada). For confocal imaging, platelets and endothelial cells were detected using a mouse monoclonal anti-CD41 (Santa Cruz, #sc-365938, Dallas, TX, USA) and a rabbit polyclonal anti-CD31 (Novus Biological, #BN100-2284, Centennial, CO, USA) primary antibodies, respectively. Positive signals were revealed with a donkey Anti-Mouse IgG H&L (Alexa Fluor^®^ 647) (Abcam, #ab150107) and donkey Anti-Rabbit IgG H&L (Alexa Fluor^®^ 555) (Abcam, #ab150074), respectively. Histology and immunochemistry images were obtained using a bright field microscope (BX45 model, Olympus, Tokyo, Japan) with QImaging QICAM Fast13 camera and Image Pro Premier software (Media Cybernetics, Rockville, MD, USA). Fluorescence images were obtained using a confocal microscope (LSM 710; Carl Zeiss, Toronto, ON, Canada). Scoring and histological procedures were performed on each of the thirteen collected graft samples as previously described [33] Ten images of histology and immunohistochemistry (200× magnification) were randomly selected in each tissue sample and a qualitative score was determined to assess vascular occlusion from these images (0, 25, 50, 75 or 100%). A 100% score means all or most vessels visualized (venules and arterioles) were occluded with neutrophils or NETs, whereas 25/50/75% scores meant that a corresponding fraction of the blood vessels were obstructed with neutrophils or NETs. All vessels visualized in the images were used for the analyses. Blinded scoring was performed by the same observer.

### 2.5. Statistics

Characteristics of donors and recipients are presented as frequency (percentages), mean ± standard deviation or median (interquartile range [IQR], quartile 1; quartile 3) as indicated. Predicted total lung capacities were calculated based on formulas defined by Mason and al. [34] using age, sex, height and weight. Inter-group differences were assessed with chi-square tests for dichotomous variables, and with Student *t* or Mann–Whitney *U* tests for continuous variables according to distribution. Random-effect longitudinal analyses were used for cellular counts and biomarkers to account for missing time-points (PROC MIXED, SAS 9.4, SAS Institute Inc., Cary, NC, USA). Estimates from the models are reported and differences at each time-point were assessed with contrasts only when the overall inter-group difference was significant. Results were considered significant if *p* < 0.05.

## 3. Results

### 3.1. Clinical Results

From June 2018 to December 2020, 172 recipients were transplanted at the CHUM. Out of these 172 recipients, 115 were consented for research (67%) but we collected all data from only 36 recipients and 25 donors. No data was collected from the remaining 79 recipients and their corresponding donors. The latter is explained by the complexity of gathering all required collaborators to collect needed data per patient to allow comparisons between groups. Nine (9) out of 36 recipients developed PGD3 within 12 h following surgery. Preoperative characteristics of recipients and their corresponding donors are presented in Table 1. None of the baseline characteristics were associated with the PGD3 group. Baseline levels of organ markers not lung-related are presented in Appendix A. When PGD grading was extended to 72 h, four of the nine (4/9) recipients with PGD3 at 12 h were still graded as PGD3. Amongst recipients graded non-PGD3 at 12 h, none developed PGD3 at 72 h.

Intra- and postoperative characteristics of patients are displayed in Table 1. Duration of surgery amongst recipients who developed PGD3 was longer (*p* = 0.044). Five recipients of the PGD3 group and three recipients of the non-PGD3 group were placed on CPB or ECMO intraoperatively (55% vs. 11%, *p* = 0.005). Postoperatively, four recipients of the PGD3 group were on ECMO while no recipient from the non-PGD3 group was on ECMO (44% vs. 0%, *p* = 0.001). Finally, PGD3 was associated with postoperative hemorrhage (*p* = 0.014) and 1-year mortality (*p* = 0.002), but prolonged (>72 h) mechanical ventilation was not statistically associated with PGD3. Postoperative levels of organ markers not lung-related are presented in Appendix A. Three hours following second lung reperfusion, PGD3 recipients had elevations in liver cytolysis biomarker (ALT, *p* = 0.008), renal dysfunction (creatinine, *p* = 0.050) and hypo-perfusion (lactic acid, *p* = 0.004).

### 3.2. NETs-Mediated Vascular Occlusion in Grafts

Tissue samples from 13 donor grafts were collected before reperfusion in recipients. From the 13 available grafts, 8 were associated with PGD3 and 5 with non-PGD3 outcome. Figure 1A–C and Figure 1D–F are representative images of lungs from both groups presenting various levels of vascular occlusion with neutrophils (MPO) and NETs (CitH3). Using a qualitative score, vascular occlusion by neutrophils changed from 20% to 38% and NETs from 15% to 34% in non-PGD3 versus PGD3 groups (*p* = 0.477 and *p* = 0.500, respectively) (Figure 1G). In blood vessels showing partial to complete occlusion, both neutrophil (MPO)-platelet (CD41) interaction and adhesion onto endothelium (CD31) were present (Appendix A). Neutrophils undergoing NETosis (CitH3) were adhered to the endothelium and present in the thrombus core (Appendix A).

### 3.3. Circulating NETs Are Associated with PGD3

NETosis kinetic was measured using three biomarkers (MPO-DNA, MPO and cfDNA), in donors and recipient samples (Figure 2A–C). For MPO-DNA, a specific NETosis biomarker, there was a significant difference over time between patients who developed PGD3 at 12 h following surgery and those who did not (*p* = 0.036). In recipients who developed PGD3 at 12 h, levels of MPO-DNA gradually increased intra- and postoperatively in the PGD3 group and were specifically elevated at 3 h following second lung engraftment, and at 48 h post-surgery. MPO levels also fluctuated significantly over time between the two groups (*p* = 0.018), being more elevated at the second lung implantation and 3 h postoperatively in the PGD3 group. Finally, cfDNA levels were also higher in the PGD3 group overtime (*p* = 0.041). Like MPO, cfDNA levels were more elevated at the second lung implantation and 3 h later, and additionally at 72 h post-surgery. For all three NETosis biomarkers, levels in donors were not significantly different between the PGD3 and non-PGD3 groups. Importantly, NETosis elevations in the PGD3 group were not due to an increase in circulating neutrophils (Figure 3A). Indeed, neutrophils and lymphocytes were similar in both groups over time (Figure 3A,B). Interestingly, we observed a relative platelet deficiency in the PGD3 group as compared with the non-PGD3 group (Figure 3C) (*p* = 0.024), and the latter was not due to differential platelet transfusion between groups (Table 1).

### 3.4. NETosis-Related Inflammation Is Associated with PGD3

Three additional inflammatory biomarkers also known to induce NETosis [35,36,37] were measured in donors and before, during and after lung transplantation in recipients (Figure 2). Preoperatively, the only biomarker associated with the PGD3 group was IL-8 (54 ng/mL vs. 20 ng/mL, *p* = 0.004). Recipient’s intra- and postoperative values of IL-8 (*p* < 0.001) and IL-6 (*p* = 0.002) were both highly associated with PGD3 development in recipients. The highest elevations of IL-8 and IL-6 were observed at 3 h following second lung engraftment, as observed with NETosis kinetics. For CRP, values in both groups were also elevated in the postoperative period, peaking at 24 h. No difference was found over time between the two groups (*p* = 0.614). Nonetheless, CRP levels dropped significantly after 48 h post-surgery in the non-PGD3 group (*p <* 0.0001), while values remained elevated in the PGD3 group. As for NETosis biomarkers, there was no significant association between the development of PGD3 in the recipient and values of IL-8, IL-6 and CRP in donors. Interestingly, independent of PGD3 development, values of CRP in donors were high while values of IL-8 and IL-6 were minimal.

### 3.5. Impact of Identified Risk Factors on NETosis

Previous studies reported that the extracorporeal circuits of CPB and ECMO are causing apoptosis, necrosis and cellular damage resulting in the release of nonspecific cfDNA in the bloodstream [38,39]. As with patients undergoing liver transplantation, cfDNA measurement may not be proportional to the levels of circulating NETs [40]. Accordingly, we observed a significant correlation between cfDNA (at 3, 48 and 72 h) and CPB/ECMO use (spearman rho = 0.51, *p* = 0.004; rho = 0.34, *p* = 0.054; rho = 0.36; *p* = 0.045). However, we did not observe any correlation between CPB/ECMO and MPO-DNA, a specific biomarker of NETosis. Other risk factors for the formation of NETs were explored and correlation assessed between postoperative NETs formation (MPO, MPO-DNA and cfDNA) and potential predictors (duration of surgery, postoperative blood loss, postoperative hemorrhage, ventilation time). Again, there was only a significant correlation between cfDNA at 24 h and postoperative blood loss (rho = 0.36, *p* = 0.045) or postoperative hemorrhage (rho = 0.42, *p* = 0.015), and cfDNA at 48 h and ventilation time (rho = 0.50, *p* = 0.005). These risk factors did not correlate with MPO-DNA nor with MPO.

In the PGD3 group, 6 recipients out of 9 were put on CPB and/or ECMO. Three of the 6 recipients had CPB/ECMO per and post-surgery, while two recipients only used CPB/ECMO during the surgery and one recipient only used ECMO in the postoperative period. Thus, 5 recipients out of 6 had CPB/ECMO intraoperatively and 4 had CPB/ECMO postoperatively. To decipher the impact of CPB/ECMO on the NETosis and inflammatory kinetics, we divided our recipients in subgroups whether they benefited or not from CPB/ECMO; thus, patients undergoing CPB/ECMO were evaluated separately from those not undergoing CPB/ECMO (Figure 4). Out of the 6 recipients who were on CPB/ECMO and developed PGD3, 4 deceased within 1 year after transplantation, whereas in the non-PGD3 group, only one (1) out of the 3 recipients who were on CPB/ECMO deceased within 1 year. Patients developing PGD3, whether or not undergoing CPB/ECMO, had higher levels of circulating IL-8 and IL-6, which peaked at 3 h postoperatively as compared to non-PGD3 recipients undergoing or not CPB/ECMO. High levels of IL-8 and IL-6 are associated with PGD3 but independent from CPB/ECMO. In addition, recipients developing PGD3 have higher levels of MPO-DNA than recipients in the non-PGD3 group, but this finding was only observed in recipients with CPB/ECMO (*p* = 0.033) in contrast to those not using CPB/ECMO (*p* = 0.735). In this PGD3 group with CPB/ECMO, elevations in MPO-DNA levels are sustained for at least 48 h after the transplantation (Figure 4G).

Based on these findings, we compared MPO-DNA levels with the four biomarkers significantly associated with PGD3 at 3 h (Figure 5). We established 2 × 2 combinations to assess the PGD3 predictive potential of these biomarkers when measured simultaneously at 3 h. Cutoffs for each biomarker were established arbitrarily at the 66th percentile to show graphically their correlation with the risk of developing PGD3. When values are over the 66th percentile for each biomarker, the risk of developing PGD3 varies from 71 to 80%. More importantly, when values are lower than the 66th percentile for each biomarker (blue zones), the risk of developing PGD3 varies from 0 to 6%. This finding is also applicable in recipients undergoing CPB/ECMO (dotted points), where the risk of developing PGD3 is minimal when values are below the 66th percentile for each biomarker (blue zones).

## 4. Discussion

Our study is the first to establish a correlation between circulating NETs and the development of PGD3 in human lung transplantation. We observed an increase in intra- and postoperative recipient levels of circulating NETosis biomarkers (MPO-DNA, MPO and cfDNA) and NETs-associated pro-inflammatory biomarkers (IL-8 and IL-6), correlating with PGD3 development. 

Neutrophils and NETs are key contributors to many occlusion-related lung complications, and their inter-relationship with platelets is highly pathogenic [41,42]. Upon their release [43], NETs can bind to endothelial cells through von Willebrand factor (vWF) [44,45] and P-selectin [35,46], providing a scaffold for the binding of platelets, neutrophils and erythrocytes, leading to fibrin deposition and thrombotic microvascular occlusion [47,48,49]. In our study, as depicted by immunofluorescence staining, the microvascular occlusion was mainly derived from neutrophils undergoing NETosis and platelet-bound neutrophils adhered onto lung endothelium. This finding was observed prior to lung reperfusion in recipients, suggesting that actual techniques to clear blood-born elements in blood vessels from donor’s lungs might be insufficient.

The local load of intravascular NETs can further be exacerbated upon reperfusion. In this study, perioperative elevations of IL-8 and IL-6, two potent inducers of NETosis and lung neutrophilic chemotaxis and migration, were associated with PGD3 [35,36,37]. These elevations could further aggravate local NETs formation after neutrophil chemotaxis in lungs. Two studies [21,23] previously correlated perioperative IL-8 and IL-6 levels with PGD3, and our study supports these findings, highlighting the early predictive potential of these biomarkers. Interestingly, we found that intraoperative levels of NETs were associated with PGD3. More precisely, biomarkers of NETosis measured directly after second lung implantation and at 3 h were closely associated with PGD3 development at 12 h. Therefore, this might suggest that circulating NETs in recipient venous blood enter pulmonary vasculature and add to the local load of intravascular NETs, perpetuating microvessel occlusion and ischemia-reperfusion lesions.

Systemic NETosis leads to circulating platelet depletion through NETs-mediated disseminated clot formation, a major contributor to microvascular hypoperfusion and multiorgan dysfunction [50]. In our study, we observed a significant decrease of circulating platelets concomitantly with circulating NETs elevation in recipients developing PGD3. These recipients also presented a slight transient elevation in multiorgan damage biomarkers, although this might be a consequence of surgery-related hemodynamic factors. Taken together, these findings support the plausibility of systemic clot formation following lung transplantation. This prothrombotic nature of NETs was also delineated in liver transplantation where cfDNA measured upon liver reperfusion correlated with coagulation activation [40]. However, further work comprising coagulation studies is needed to confirm these exploratory findings in the field of lung transplantation.

CPB and ECMO can increase circulating levels of NETs and proinflammatory cytokines [38,51,52]. Thus, we performed subgroup analyses to discern the role of CPB/ECMO on PGD3 development in our cohort. Recipients developing PGD3, whether or not undergoing CPB/ECMO, have higher levels of circulating IL-8 and IL-6 that peak at 3 h after surgery as compared to non-PGD3 recipients. This suggests that high levels of IL-8 and IL-6 in PGD3 recipients are independent from CPB/ECMO procedures. These findings could be explained by the fact that circulating cells from these PGD3 recipients have higher intracellular IL-8 and IL-6 contents or are more primed to synthesize and release both cytokines. Interestingly, we observed that PGD3 vs. non-PGD3 recipients undergoing CPB/ECMO present a significant higher level of circulating NETs (MPO-DNA), which was maintained for at least the first 48 h after transplantation. These data demonstrate that CPB/ECMO procedures contribute to NETosis in patients developing PGD3 and correlate with high circulating levels of IL-8 and IL-6, which are both inducers of NETosis. This later observation suggests that patients undergoing CPB/ECMO procedures with elevated levels of IL-8, IL-6 and MPO-DNA at 3 h after transplantation are at higher risk of developing PGD3 and its associated complications; 4 out of 6 of these recipients were deceased within 1 year. Thus, the measurement of preoperative IL-8 as well as IL-8, IL-6 and MPO-DNA at 3 h post-transplantation, peculiarly for recipients undergoing CPB/ECMO intraoperatively, might provide valuable prognostic information.

Previous studies reported clinical benefits of grading PGD at ICU arrival (0hr), 6 h and 12 h [28,29,30,53]. In our study, all recipients with PGD3 at 72 h were rightly classified as PGD3 as early as at 12 h, therefore, we elected to classify recipients developing PGD3 at 12 h. Moreover, perioperative measurement of NETosis and inflammatory biomarkers correlated with PGD3 at 12 h. Therefore, the earliest measurement of NETosis and inflammatory biomarkers could be useful to rapidly identify recipients at high vs. low risks of PGD3, facilitating the decision to potentially treat the recipients with therapeutic strategies targeting NETosis and inflammation.

NETs measurements in bronchoalveolar lavage [10,54] and in EVLP perfusates [27] are important factors to predict recipient outcome. Considering the difficult task of predicting recipients’ clinical progression, a predictive algorithm based on early measurement of circulating biomarkers (3 h post-transplant) was illustrated to provide tangible data to discriminate patients at high vs. low risk of developing PGD3. When recipients have high levels of either one or more biomarkers at 3 h post-transplantation (i.e., they are in the intermediate [white] or high risk [red] zones), the risk of developing PGD3 reaches up to 80% in our cohort. On the other hand, when biomarkers’ levels are low at 3 h post-transplantation (i.e., they are in the low risk [blue] zones), the risk of developing PGD3 is minimal, even in recipients undergoing CPB/ECMO. This algorithm could provide informative data whether recipients should benefit from a treatment with selective inhibitors of NETs synthesis and/or with DNAse I to degrade DNA from NETs complexes. Further research should combine the measurement of NETosis biomarkers in the blood and/or perfusates with clinical parameters to establish a robust score for PGD3 prediction.

Finally, severe PGD is associated with a deficiency in intra-bronchial DNase I [55] and that intra-bronchial DNase I therapy can reduce lung injury following murine lung transplantation [10]. In a sepsis animal model, DNase I therapy given intraperitoneally (IP) reduced organ damage and improved survival [56,57]. DNase I (IP or intravenously, IV) was also beneficial in IRI involving kidneys, heart and liver [58,59,60]. In a phase randomized 1b, placebo-controlled trial of 17 patients with systemic lupus erythematosus (SLE), it was shown that intravenous DNase I was safe and well tolerated [61]. Specifically, DNase I provided sustained catalytic activity up to 8 h in these patients [61]. There were no significant adverse events following administration, and treatment was not associated with the development of neutralizing antibodies against recombinant DNase I [61]. Thus, IV DNase I administration could prevent NETs-mediated complications following human lung transplantation.

### Study Limitations

The plurality of blood collection time points made it technically difficult to achieve the complete acquisition of all blood samples. Random-effect longitudinal analyses were used to account for missing values and this accommodation method increased the statistical strength, but we were unable to assess the predictive potential of our biomarkers statistically due to low sample size. The low number of patients also hindered us from adjusting our results with cofounding variables. In this study, all ECMO were considered having PGD3 since ECMO was used to support patient ventilation/oxygenation and not for hemodynamic support. As shown with CPB, the use of ECMO likely activates neutrophils to produce NETs [38]. Therefore, it is difficult to dissociate the implication of ECMO from NETosis in PGD3 development, and the prognostic value of NETs in PGD needs further evaluation. However, even with small sample size due to the exploratory nature of the study, subgroup analyses with or without CPB/ECMO were accordingly presented to enhance the comprehension of CPB/ECMO on our cohort. Additionally, well-described characteristics associated with PGD3 such as donor-recipient allograft size mismatch, recipient sex, recipient body mass index, prolonged mechanical ventilation and others were not associated with PGD3, and this might be a consequence for the lack of power [4]. In our study, we elected to classify recipients developing PGD3 at 12 h. However, in the 2016 ISHLT guidelines [2], it is suggested that PGD3 present at later times (48 and 72 h) appears to have the greatest effect on long-term outcomes, including mortality.

## 5. Conclusions

In conclusion, this study linked high circulating NETs in recipients with a higher risk to develop PGD3 following human lung transplantation. The perioperative dosage of IL-8, IL-6 and NETosis biomarkers (MPO-DNA, MPO and cfDNA), particularly in recipients undergoing CPB/ECMO, could lead to the development of specific diagnostic tools in order to identify recipients at high risk of developing PGD3.

## Figures and Tables

**Figure 1 cells-11-03420-f001:**
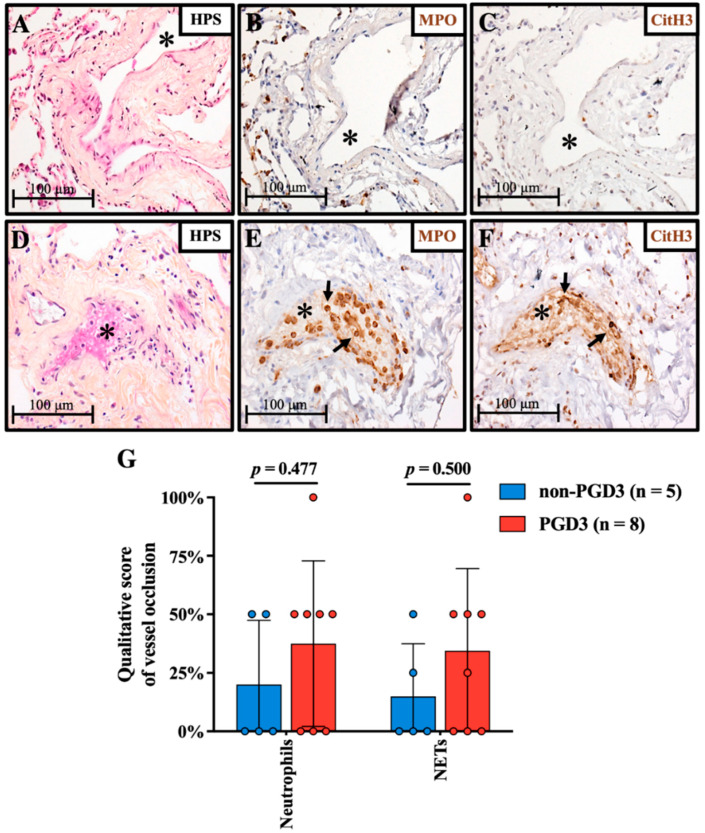
Neutrophil and NETs-mediated vessel occlusion in grafts prior to transplant. Morphology was assessed by HPS staining. Vascular occlusion occurs to varying degrees, some arterioles or venules (*) are not occluded (**A**–**C**) while others are completely occluded (**D**). Vascular occlusion mainly originates from neutrophils ((**E**); MPO; black arrows) and activated neutrophils undergoing NETosis ((**F**); CitH3; black arrows). Qualitative assessment (**G**) of vessel occlusion in grafts which led to PGD3 (n = 8) as compared with non-PGD3 (n = 5). Each graft obtained was scored qualitatively for vascular occlusion (0, 25, 50, 75 or 100%). A 100% score means all or most vessels visualized (venules and arterioles) were occluded with neutrophils or NETs, whereas 25/50/75% scores mean about 25/50/75% of the blood vessels were obstructed with neutrophils or NETs. Means and standard deviations are represented with bars and individual scores for each graft are plotted with colored dots (Student *t* test). Original magnification, ×200.

**Figure 2 cells-11-03420-f002:**
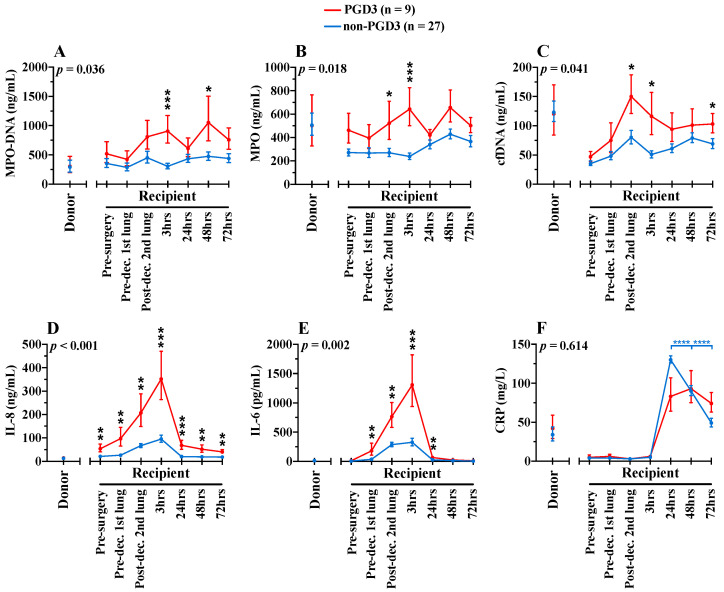
Pre-, intra- and postoperative serum levels of NETosis (**A**–**C**) and inflammatory (**D**–**F**) bi-omarkers in donors (n = 25) and recipients (n = 36). Blue lines show biomarkers in non-PGD3 recipients (n = 27) and red lines in PGD3 recipients (n = 9) (Mixed-method analyses). The 95% confidence intervals are represented with bars. *p* values represent the global inter-group comparisons over time. Significant time-point differences are illustrated with asterisks (*). * *p* < 0.05, ** *p* < 0.01, *** *p* < 0.001 and **** *p* < 0.0001.

**Figure 3 cells-11-03420-f003:**
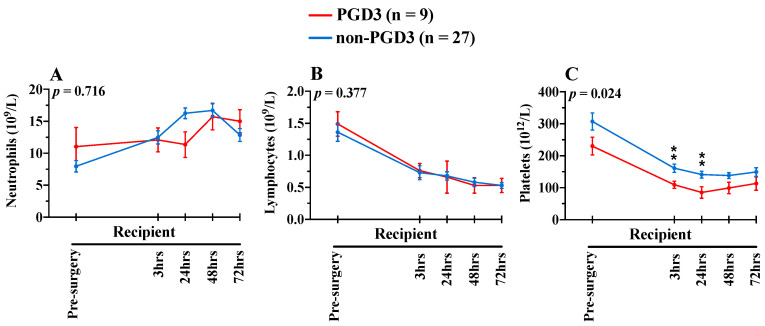
Pre- and postoperative circulating neutrophils (**A**), lymphocytes (**B**) and platelets (**C**) in recipients (n = 36). Blue lines show biomarkers in non-PGD3 recipients (n = 27) and red lines in PGD3 recipients (n = 9) (Mixed-method analyses). The 95% confidence intervals are represented with bars. *p* values represent the global inter-group comparisons over time. Significant time-point differences are illustrated with asterisks (*). ** *p* < 0.010.

**Figure 4 cells-11-03420-f004:**
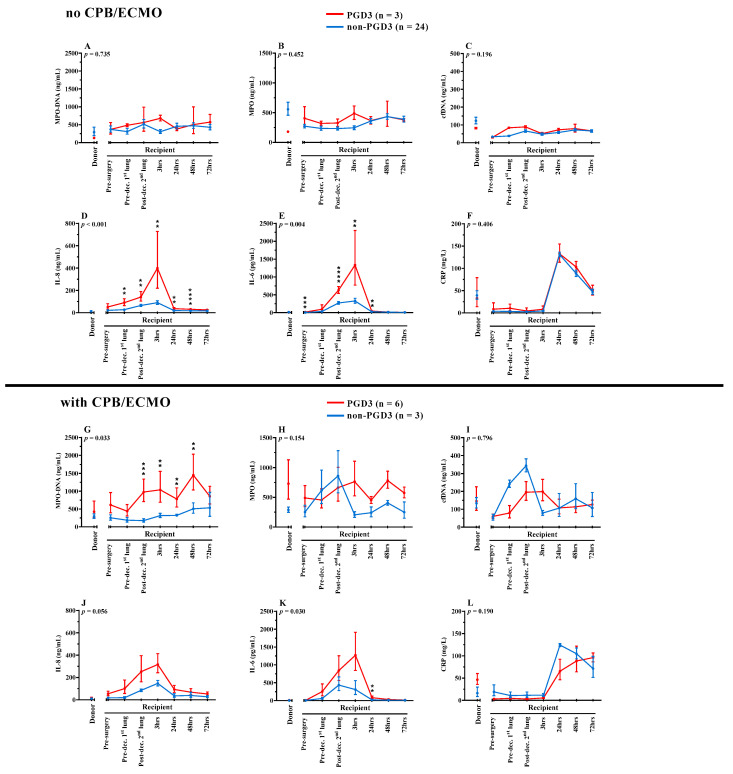
Serum levels of NETosis and inflammatory biomarkers in recipients without CPB/ECMO (**A**–**F**) (n = 27) or with CPB/ECMO (**G**–**L**) (n = 9). Donors (n = 25) and recipients’ (n = 36) pre-, intra- and postoperative serum levels of NETosis (**A**–**C**,**G**–**I**) and inflammatory (**D**–**F**,**J**–**L**) biomarkers are displayed. Blue lines show biomarkers in non-PGD3 recipients and red lines in PGD3 recipients (Mixed-method analyses). The 95% confidence intervals are represented with bars. *p* values represent the global inter-group comparisons over time. Significant time-point differences are illustrated with asterisks (*). ** *p* < 0.01, *** *p* < 0.001, **** *p* < 0.0001.

**Figure 5 cells-11-03420-f005:**
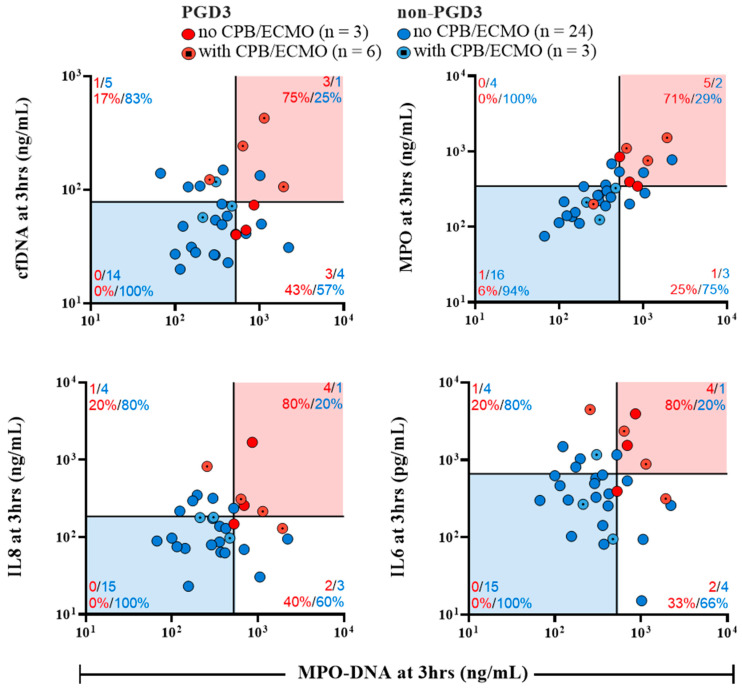
Prediction of PGD3 onset based on early postoperative (3 h) values of NETosis biomarkers and inflammatory cytokines inducing NETosis. Cut-off lines represent the 66th percentile value for each biomarker and limit four quadrants. The number of recipients belonging in each quadrant is displayed with colored numbers. For every binary combination of biomarkers, recipients located in the red zones have the highest risk of developing PGD3 whereas recipients in the blue zones have the lowest risk of developing PGD3 at 12 h. Both X and Y axis have logarithmic scales.

**Table 1 cells-11-03420-t001:** Pre-, intra- and postoperative characteristics of donors and recipients undergoing bilateral lung transplantation.

	All Patients (n = 36)	Non-PGD3 (n = 27)	PGD3 (n = 9)	*p* Value
** * Donor characteristics * **				
Male	18 (50%)	14 (52%)	4 (44%)	1.000
Age (years)	45 ± 17	43 ± 16	50 ± 15	0.281
Body mass index (kg/m^2^)	26 (23–29)	25 (23–30)	25 (25–27)	0.889
Cause of death				0.766
- Trauma	15 (43%)	12 (44%)	3 (33%)	
- Stroke	12 (33%)	9 (33%)	3 (33%)	
- Suicide	9 (25%)	6 (22%)	3 (33%)	
Type of organ donation				0.392
Donation after brain death	26 (72%)	18 (67%)	8 (89%)	
Donation after circulatory death	10 (28%)	9 (33%)	1 (11%)	
Duration of cold ischemia (minutes)	242 (137–278)	237 (137–278)	263 (177–287)	0.406
pTLC donor-to-recipient ratio	1.02 (0.85–1.09)	1.03 (0.85–1.09)	0.96 (0.84–1.06)	0.618
PaO_2_/FiO_2_ ratio prior to organ harvest (mmHg)	441 ± 73	448 ± 60	418 ± 105	0.436
White blood cells (10^9^/L)	13 (11–18)	13 (11–16)	18 (12–24)	0.107
** * Recipient characteristics * **				
Male	28 (78%)	22 (82%)	6 (67%)	0.352
Age (years)	62 (53–64)	62 (46–64)	62 (58–66)	0.368
Body mass index (kg/m^2^)	24 ± 4	24 ± 4	25 ± 5	0.382
pTLC (L)	6.5 (5.6–6.9)	6.7 (6.1–7.0)	6.3 (5.3–6.4)	0.136
Diabetes	14 (39%)	9 (33%)	5 (56%)	0.194
Smoking				0.431
Never	14 (39%)	12 (44%)	2 (22%)	
Former	21 (58%)	15 (56%)	6 (67%)	
Lung disease				0.160
Pulmonary hypertension	2 (6%)	0 (0%)	2 (22%)	
Idiopathic lung fibrosis	12 (33%)	10 (37%)	2 (22%)	
Cystic fibrosis	5 (14%)	4 (15%)	1 (11%)	
Chronic obstructive pulmonary disease	12 (33%)	9 (33%)	3 (33%)	
Interstitial pulmonary disease	5 (14%)	4 (15%)	1 (11%)	
White blood cells (10^9^/L)	10 (7–15)	10 (7–11)	12 (7–15)	0.355
** * Intraoperative characteristics * **				
Duration of surgery (minutes)	250 ± 51	240 ± 43	280 ± 63	0.044
EVLP	0 (0%)	0 (0%)	0 (0%)	1.000
CPB/ECMO	8 (22%)	3 (11%)	5 (55%)	0.005
Red blood cells transfusion	28 (78%)	21 (78%)	6 (67%)	0.505
Platelets transfusion	8 (22%)	4 (15%)	4 (44%)	0.064
Volume of blood loss (liters)	1.2 (1.0–2.0)	1.2 (1.0–1.5)	1.5 (1.5–3.0)	0.101
** * Postoperative characteristics * **				
Postoperative hemorrhage	4 (11%)	1 (4%)	3 (33%)	0.014
ECMO	4 (11%)	0 (0%)	4 (44%)	0.001
Mechanical ventilation > 72 h	8 (22%)	4 (11%)	4 (44%)	0.064
PGD3 at 72 h	4 (11%)	0 (0%)	4 (44%)	0.001
Death in first year following surgery	5 (13%)	1 (4%)	4 (44%)	0.002

Data are presented as frequencies (%), means ± standard deviation or medians (IQR) where appropriate. CPB, cardio-pulmonary bypass; ECMO, extracorporeal membrane oxygenation; EVLP, ex vivo lung perfusion; pTLC, predicted total lung capacity.

## Data Availability

Data may be made available from the corresponding authors upon reasonable request.

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
