# Peer review of "Correlation between Neutrophil Extracellular Traps (NETs) Expression and Primary Graft Dysfunction Following Human Lung Transplantation"

_cells, 2022, doi:10.3390/cells11213420_

Round 1
Reviewer 1 Report
Authors tried to correlate NETosis with recipient PGD3 development in a prospective exploratory study. The finding is interesting. There are some suggestions as the following.
1. The title may not be precise, “Impact of neutrophil extracellular traps (NETs) in primary graft dysfunction following human lung transplantation”, however, this study only found the possible correlation between NETs and PGD, with no further exploration about how NETs impacted PGD.
2. As a prospective observational study, this study should divide the patients into two groups according to the level of NETs, so as to explore whether the difference in NETs correlated with later PGD. However, the tables and figures in this article indicated that this was a retrospective study.
3. Table1 told us that the duration of surgery, application of CPB/ECMO, postoperative hemorrhage differed greatly between PGD and non-PGD group, which were the top risk factors to PGD. To reduce bias, this study may explore whether those risk factors contributed to the formation of NETs.
4. The number of patients enrolled in this study is inadequate. Given the fact that diseases like Pulmonary hypertension, COPD, IPF and CF, the top indications for transplant, are of high heterogeneity, which may contribute to high risk of bias, I suggest Propensity Score Match for further research.
5. In the section 3.2, the author explored the formation of NETs in donor lung graft prior to transplant and found no difference between PGD and non-PGD group. However, the NETs formation in donor lungs were displayed in figure 2 as a major composition of the article, which did not relate closely to the main topic. I suggest moving this figure to supplemental material.
6. In the section 3.3, “a relative platelet deficiency in the PGD3 group as compared with the non‐PGD3 group, and the latter was not due to differential platelet transfusion between groups ”. This point is important, I suggest showing it in the main text.
7. In the section 3.5, 6 recipients out of 9 were put on CPB and/or ECMO per or post surgery in the PGD3 group. But table 1 showed that 5 in operation and 4 post operation put on CPB and/or ECMO. This inconsistency may raise readers confusions, please give a detailed description.
8. Figure 4 seems telling us that ECMO contributed to the formation of NETs. However, ECMO is critical for PGD. These factors complicated the relationships of NETs, ECMO and PGD. The prognostic value of NETs in PGD needs further evaluation.
9. In figure 5, please explain why 66th percentile was set to evaluate the risk of biomarkers.
10. There are some recent relevant literature (2020-2022) that authors should include in the introduction and discussion parts, including but not limited to
(1) Mitochondrial DNA Stimulates TLR9-Dependent Neutrophil Extracellular Trap Formation in Primary Graft Dysfunction. Am J Respir Cell Mol Biol. 2020 Mar; 62(3):364-372.
(2) Long non-coding RNA X-inactive specific transcript silencing ameliorates primary graft dysfunction following lung transplantation through microRNA-21-dependent mechanism. EBioMedicine. 2020 Feb; 52:102600.
Reviewer 2 Report
Thank you for giving me a great opportunity to review the article entitled “Impact of neutrophil extracellular traps (NETs) in primary graft dysfunction following human lung transplantation”. I have read the manuscript with much interest as I expect this field has a variety of therapeutic targets in lung transplantation. I hope my comments below make the manuscript even better.
1) Please indicate the groups and the number of each group in the abstract.
2) Were all the patients received double-lung? I’d assume all the cases were performed double-lung transplants but please specify in Table 1.
3) In my understanding, the PGD score at 72 hrs, not 12 hrs, is the most critical point to predict both short-term and long-term outcomes. Although the authors described that the PGD score at 12 hrs still might work in material and methods, please include this point in the study limitations and discussions.
4) In the study period, how many cases of lung transplantation were performed in total in your institution? And what is the percentage of the cases included in this study?
5) In the entire cohort, 5 patients out of 36 cases (13%) died in the first 120 days following surgery. I believe this number is relatively higher than the expected number. From this point, I’d recommend including the LAS score to show the recipient disease severity in your cohort. Also, please describe the reason for higher death rates in the cohort.
6) Please show the PGD score at 0, 24, and 48 hrs also in Table 1.
7) Please include the smoking history of the donor in Table1.
8) I believe, in the manuscript, the critical point/figure would be Figure 1G and Figure 3C. However, the data in Figure 1G did not show a significant difference between the groups with a limited number, while Figure 1C showed a significant difference between the groups. This result seems a bit wired for me because the number of PGD3 at 72 hrs was just 4 cases in the PGD3 group but still had higher levels of cfDNA at 72 hrs. Do you have any comments on that?
9) Line 365: Sayah and al: typo? Please correct it.
10) Reference #10 showed that cfDNA levels in BAL were an important factor to predict graft survival in clinical transplantation. I’d suggest including this point in the manuscript to clarify the novelty in your study.
11) In Figure 4, X-axis has 0.75, 2,,,,8 and other categorical values. Also, they are overlapped and confusing. Please correct it nicely.
12) In Figure 5, are these figures expressed in logarithms? Please indicate it in the figure legend and figure itself.
13) Finally, I believe NETs in ex vivo lung perfusion (Ref #27) would be strongly correlated with this study to explore potential biomarkers and therapeutics. Would you please include this point to indicate similarities with this previous study and what needs to be further researched in the discussion?
Round 2
Reviewer 1 Report
All of my comments have been well addressed. No further comments.
Reviewer 2 Report
Thank you for giving me another opportunity to review the manuscript.
The biggest concern about this study/manuscript is its study design as I indicated in my first comments. Although the author described it honestly in the text, I don't feel like it's an appropriate study after my careful reviewing of revised ones. This is because the study was conducted in a prospective way with a good number of consented cases yet the final corrected samples are too limited. From this point, I suggest including or adding more cases and re-analyzing the data.
